# The Association between Change in Lifestyle Behaviors and Mental Health Indicators in Immunosuppressed Individuals during the COVID-19 Pandemic

**DOI:** 10.3390/ijerph20032099

**Published:** 2023-01-23

**Authors:** Tara Zeitoun, Audrey Plante, Catherine M. Sabiston, Mélanie Dieudé, Isabelle Doré

**Affiliations:** 1Department of Nutritional Sciences, Temerty Faculty of Medicine, University of Toronto, Toronto, ON M5S 1A8, Canada; 2CHUM Research Centre, Montréal, QC H2X 0A9, Canada; 3Faculty of Kinesiology and Physical Education, University of Toronto, Toronto, ON M5S 2W6, Canada; 4Department of Microbiology, Infectiology and Immunology, Faculty of medicine, Université de Montréal, Montréal, QC H3C 3J7, Canada; 5Canadian Donation and Transplant Research Program (CDTRP), Edmonton, AB T6G 2E1, Canada; 6Héma-Québec, Montréal, QC H4R 2W7, Canada; 7School of Kinesiology and Physical Activity Sciences, Faculty of Medicine, Université de Montréal, Montréal, QC H3T 1J4, Canada; 8Social and Preventive Medicine Department, School of Public Health, Université de Montréal, Montréal, QC H3N 1X9, Canada

**Keywords:** COVID-19 pandemic, mental health, physical activity, sleep duration, sedentary behavior, immunosuppressed population

## Abstract

Little is known on how changes in lifestyle behaviors affect mental health among immunosuppressed individuals who observed stricter physical and social distancing measures due to higher risk of complications during the COVID-19 pandemic. This study examines the association between changes in moderate-to-vigorous physical activity (MVPA), sedentary time (ST) and sleep duration following COVID-19 outbreak on mental health indicators of immunosuppressed individuals and their relatives. Participants (*n* = 132) completed an online questionnaire between May and August 2020. Linear regressions were conducted to assess the associations between an increase or decrease in lifestyle behaviors and mental health indicators. Individuals with decreased MVPA and increased ST experienced higher distress, anxiety and depressive symptoms. Those who reported an increase or decrease in sleep had higher levels of stress, distress and depressive symptoms. Decreases in sleep was associated with higher anxiety symptoms. Lifestyle behaviors in the context of a stressful life event such as the COVID-19 pandemic may impact mental health indicators of immunosuppressed individuals and their relatives.

## 1. Introduction

A healthy lifestyle includes a range of activities to encourage habits that reduce the risk of morbidity and increase life expectancy. Such a healthy lifestyle is encompassed by a plethora of habits such as, but not limited to, healthy eating, socialization, regular physical activity and adequate sleep [1]. Regular physical activity is associated with many benefits for physical and psychological health [2,3]. Physical activity is a tool shown to be effective in reducing anxiety [4,5] and depressive symptoms [4], stress [6] and fatigue [7,8], not only in the general population, but also in immunosuppressed individuals [9,10]. Unhealthy changes in lifestyle behaviors (e.g., decreased physical activity, increased sedentary behavior, reduction of quality and quantity of sleep) related to stressful life events could increase stress and anxiety [11,12,13]. As such, encouraging healthy lifestyles may be an effective way to better protect mental health and promote resilience when individuals are faced with stressful life events. 

The instigation of physical and social distancing (PSD) measures related to the COVID-19 pandemic [14] has extensively impacted the mental health of most individuals worldwide [15]. COVID-19 PSD related measures have also posed an obstacle to maintaining healthy lifestyles due to closures of physical activity spaces (e.g., sports centers, recreation facilities, national parks, playgrounds), quarantining for those that may have been in contact with COVID-19, mandatory stay-at-home policies [12,16] and fear of going outside the house for some individuals. The immunosuppressed population’s mental health and lifestyle behaviors are likely to be disproportionately impacted by the PSD measures related to the COVD-19 pandemic. Indeed, immunocompromised individuals are at an increased risk of severe complications due to COVID-19 compared to the general population [17,18], which may make them feel more vulnerable, worried and isolated. Individuals who are immunocompromised are more likely to contract illnesses or be ill for extended periods of times due to their weakened immune system. To protect their loved ones, immunosuppressed patients’ close relatives (parents, guardians, children, siblings, partners) also tend to adopt stricter PSD measures. The physical, mental and social well-being of home care providers in the context of the pandemic has been largely overlooked; specific resources and interventions for people with chronic conditions in the home context need to be considered to support informal home care providers [19,20,21]. Consequently, immunosuppressed individuals and their relatives are a high-risk group in terms of mental health in the current COVID-19 context [22].

In this framework, maintaining healthy lifestyles become a major challenge, which translates to reduced physical activity, increased sedentary behaviors and poor sleep quality [11,23,24,25]. Benefits of physical activity on mental health have been examined in the scope of stressful life events such as the COVID-19 pandemic [16,26]. Results from studies in Canada and other countries with similar PSD measures found that individuals who were more physically active had better mental health [11,12]. A Canadian study revealed that adult participants engage in physical activity as a coping strategy for mental health due to COVID-19 [27]. Alternatively, individuals who decreased their physical activity reported higher feelings of stress, loneliness and depressive symptoms [11] and those who were inactive had higher anxiety levels compared to those who were active [12]. With decreased physical activity and compulsory confinement, sedentary behaviors (e.g., recreational screen time, active screen time b when working, lack of commuting) were observed to increase during the COVID-19 pandemic [28,29]. Recent studies suggest that increased sedentary time is also associated with more depressive or anxiety symptoms [13]. 

Sleep duration is another lifestyle behavior that is vital to mental health and disturbances in sleep quality effect mental and physical performance. Lack of sleep can impair psychological functioning and decision making, jeopardize immune response, increase accidents, lead to mood changes and increase medical expenditures [30]. Moreover, insufficient sleep and poor sleep quality are associated with more depressive and/or anxiety symptoms [31]. Though the relationship between lifestyle factors such as physical activity, sedentary time and sleep time and overall mental health is understood, the effects of a drastic change in lifestyle, as seen in the COVID-19 pandemic, on mental health in vulnerable groups remains equivocal. 

It is unknown whether the benefits of physical activity on mental health observed during COVID-19 in the general population extend to immunosuppressed individuals and their relatives. As this population is likely to adopt strict PSD measures due to their immunosuppression status, maintaining physical activity can represent an even greater challenge compared to the general population. Considering the high level of stress immunosuppressed individuals experience while avoiding illness, it is crucial to assess mental health in a vulnerable population. Examining the relationship between rapid lifestyle changes and mental health outcomes in an immunosuppressed population may help prepare and prevent unwanted mental health outcomes during other similar stressful lifestyle events. 

This study aims to comprehensively investigate the association between changes in lifestyle behaviors and mental health indicators in an immunosuppressed population and their relatives in the context of COVID-19 related stressful life events. We specifically aimed to assess the association between changes in moderate-to-vigorous physical activity (MVPA), sedentary time and sleep duration before and following COVID-19 outbreak on mental health indicators of stress, distress, resilience, anxiety and depressive symptoms. 

## 2. Methods 

### 2.1. Study Design

Our team conducted the COVID-Immuno Study, a cross-sectional study aimed to assess the impact of the COVID-19 pandemic PSD measures on lifestyle behaviors and mental health in an immunosuppressed population and their relatives. Data were collected by our research team from online self-reported questionnaires between May and August 2020. The study received ethics approval from the Ethics Research Committee of the Centre de Recherche du Centre Hospitalier de l’Université de Montréal. 

### 2.2. Data Collection

Participants were recruited through partner organization websites or social media (Canadian Donation and Transplantation Research Program (CDTRP), Kidney Foundation of Canada (KFC), Canadian Transplant Association (CTA) and Cystic Fibrosis Canada (CFC)). The inclusion criteria included: (i) age 15 or older; (ii) English or French-speaking; (iii) being transplanted and immunosuppressed, being immunosuppressed for other reasons, being a close relative (spouse, child, or parent) to an immunosuppressed individual or a donor (iv) having access to the internet. In specific, participants were asked to identify if they were either, (i) a transplant, tissue, or stem cell recipient; (ii) a family member/relative of a transplant/tissue/stem cell recipient or donor; (iii) an organ, tissue, or stem cell donor; (iv) none of the above; or, (v) prefer not to answer. Individuals who selected “none of the above” had an option of explaining if they were immunosuppressed for other reasons, but not transplanted. All eligible participants were invited to complete an online consent form and questionnaire in English or French. 

### 2.3. Sociodemographic Variables

Participants were asked to provide socio-demographic information, including age and employment status and clinical information such as COVID diagnosis, chronic physical health conditions (e.g., osteoarthritis, diabetes, high blood pressure), transplant status and medication usage. 

### 2.4. Mental Health Indicators

Stress levels were captured using a question frequently used in various large surveys such as the Canadian Community Health Survey (CCHS) [32]. Participants were asked to report on a 5-point scale ranging from 1 (not stressful at all) to 5 (extremely stressful) of how stressful most of their days were before and after the COVID-19 pandemic. 

Distress was assessed by asking participants to select on a scale of 0 to 100 (0 = “No distress”; 100 = “Extreme distress”) how much distress they had been experiencing over the past week. Similar Subjective Units of Distress Scale (SUDS) have been used extensively to measure the perceived intensity of distress [33].

Anxiety symptoms were measured using the Generalized Anxiety Disorder 7-item scale (GAD-7) [34]. Participants were asked how often, during the last week, they were bothered by each symptom (e.g., feeling nervous, being restless) on a 4-point scale (0 = not at all, 4 = nearly every day), with higher scores reflecting a more frequent experience of anxiety emotions. Internal consistency [35] of the scores in this sample was 0.89. Evidence of construct validity among the general population has been reported previously [36]. This questionnaire has also been reported to be a suitable screening instrument in more vulnerable settings such as pregnancy and with individuals in primary health care clinics [37,38]. 

Depressive symptoms were measured using the Depressive Symptoms Scale, a 9-item scale based on the Patient Health Questionnaire-9 (PHQ-9) [39]. Participants were asked how often they were bothered by nine specific emotions (e.g., feeling down, feeling tired) relevant for assessing depression over the past week on a 4-point scale (0 = not at all, 4 = nearly every day), with higher scores reflecting a higher frequency of experiencing depressive symptoms. Internal consistency [35] of the scores in this sample was 0.85. Evidence of construct validity has been reported previously in vulnerable populations and meta-analyses [40,41]. 

Resilience, defined as the ability to recover from stress or to bounce back, was captured using the 6-item Brief Resilience Scale (BRS) [42]. Participants were asked to report on a 5-point scale (1 = strongly disagree, 5 = strongly agree) how much they agree or disagree with statements related to resilience (e.g., I usually come through difficult times with little trouble), with higher scores reflecting a stronger frequency of feeling resilient. Internal consistency [35] of the scores in this sample was 0.85. Evidence of construct validity has been reported previously [43,44]. 

### 2.5. Lifestyle Behaviors

Informed by the International Physical Activity Questionnaire (IPAQ) [45] and other valid and reliable measures of physical activity [46], items were developed for the current study. Participants self-reported days of MVPA for two specific periods, before COVID-19 and in the past 7 days. To help participants define the inception of the COVID-19 pandemic, the following information was provided: “The next questions asks about your physical activity, sedentary time and sleep. For each question, we will ask you to think about a typical week before the COVID-19* pandemic and the past 7 days (*COVID-19 was declared a global pandemic on 11 March 2020)”.

**Moderate-to-vigorous physical activity** (MVPA) was measured by asking participants, “How many days per week did you do vigorous aerobic activity, such as aerobics or fast bicycling?”, and “How many days per week did you do moderate aerobic activity, such as jogging, bicycling at a regular pace, or doubles tennis?”, respectively. They were asked to report each activity frequency for two reference periods: before COVID-19 and over the past 7 days. The average number of days spent in moderate-vigorous physical activity was calculated for both before COVID-19 and for the last 7 days by summing the average number of days spent at each of the two intensities. Change in physical activity was categorized nominally as decreased whenever a decline of at least one day in the average number of days in MVPA was observed, as stable when no change was observed and as increased when an increase of at least one day was observed. We utilized one day as the unit of change as this was the smallest unit in the response option. 

**Sedentary time** was assessed similarly to MVPA. Participants were asked to report the average number of hours and minutes spent in sedentary positions (e.g., sitting, lying down) on a typical weekday and on a weekend day on a typical week before COVID-19 and in the past seven days, respectively. Sedentary time averages for a typical week prior to COVID-19 and for the last seven days were calculated with the following formula: ((no. of minutes of sedentary time on weekdays) × 5 + (no. of minutes of sedentary time on weekend days) × 2)/7. We categorized individuals nominally by their pre- and post-COVID-19 change in sedentary time into either decreased (decrease of ≥10 min), increased (increase of ≥10 min) or stable (decrease or increased of <10 min).For measurement, 10-minute bouts of change for sedentary time were included to capture perceived change, based on previous literature assessing changes in MVPA bouts [47]. 

**Sleep duration** before and after the onset of COVID-19 were measured by asking the participants to report the average amount of hours and minutes slept on a typical weekday and on a weekend day in a typical week before COVID-19 and in the past seven days, respectively. Sleep time averages for a typical week prior to COVID-19 and for the last seven days were calculated with the following formula: ((no. of minutes sleeping on weeknights) × 5 + (no. of minutes sleeping on weekend nights) × 2)/7. We categorized individuals nominally by their pre- and post-COVID-19 change in sleep duration to either decreased (decrease of ≥10 min), increased (increase of ≥10 min) or stable (decrease or increase of <10 min) and 10-minute bouts of change for sleep duration were included to capture perceived change, based on previous literature assessing changes in MVPA bouts [47].

### 2.6. Statistical Analyses

Preliminary analyses included descriptive analyses to assess distributions, identify outliers, compute proportions, means and standard deviations. Descriptive analyses were also conducted to compare each mental health indicator (stress, distress, resilience, anxiety, and depression symptoms) according to lifestyle behavior (moderate-vigorous physical activity, sedentary time and sleep duration over the past 7 days) and change categories (increase, decrease, stable). 

Multivariate linear regressions were used to assess whether an increase or decrease in lifestyle behaviors MVPA, sedentary time and sleep duration is associated with each mental health indicator, for stress, distress, resilience, anxiety, and depressive symptoms, compared to no changes. Linear regressions were chosen to estimate the association between the categorical exposure of lifestyle behavior changes and continuous outcome of mental health indicators. Separate models were conducted for each lifestyle behavior with each specific mental health indicator. Models were adjusted for potential confounders identified in the literature, which include age [48], sex [49,50] and physical diagnosis [51,52]. All covariates were determined a priori and chosen based on theoretical grounds and known associations with the exposure and outcome of interest. All analysis was conducted using R (version 4.2.0) [53] and the “rms” package (version 6.1) [54].

## 3. Results

Participants’ characteristics are presented in Table 1. After excluding individuals with missing data (*n* = 5), 132 participants (65% females) remained in the analytical sample. The majority of the participants were aged between 35 and 69 years old (77%). Most of the participants (76%) were either a transplant, tissue, or stem cell recipient, 18% of the participants were relatives of a transplant/tissue/stem donor, 3% of the population were organ, tissue or stem cell donors and 3% were immunosuppressed for other undisclosed reasons. There were no significant differences in the reporting of lifestyle behaviors, sedentary time, sleep duration and MVPA, between participant groups. A majority of participants were not on anxiolytic or anti-depressant medications (80%). Participants reported decreasing their MVPA by an average of 1 day/week (SD = 3.1 days/week) and increasing their sedentary time by an average of 106.8 min per day (SD = 131.6 min/day) since the onset of the COVID-19 pandemic. Participants reported, on average, a minimal decrease in sleep time (mean (SD) = −1.4 min/day (SD = 80.7 min/day)). When grouped in lifestyle behaviors change categories, we found that 48% of participants reported stable MVPA, followed by 33% who decreased their MVPA and 19% who reported increasing their MVPA. Most of the participants reported an increased in sedentary time (68%), while only 9% decreased and 23% remained stable. In our sample, 39% increased their sleep duration, 33% decreased and 28% remained stable. 

Means and standard deviations for each mental health outcome according to lifestyle behavior change groups are reported in Table 2. These descriptive statistics suggest that participants who decrease MVPA report higher level of stress, distress anxiety and depressive symptoms compared to those who remain stable or increase MVPA. Participants who increase sedentary time report higher distress, anxiety and depressive symptoms compared to those who remain stable or decrease sedentary time. For sleep, participants who remain stable report lower stress level, distress, anxiety and depressive symptoms compared to those who report changes in their sleeping habits, whether these increased or decreased. Detailed measures for lifestyle behavior (MVPA, sleep duration and sedentary time) before COVID and in the past 7 days, and different means and standard deviations were extracted per change category (decrease, stable, increase) and are presented in Appendix A. 

Results from the linear regressions (Table 3) show that, compared to participants for whom MVPA remain stable, participants who decreased MVPA report higher distress ($\hat{\beta }$(95% CI) = 9.09 (0.52, 17.65)), anxiety symptoms ($\hat{\beta }$(95% CI) = 1.89 (0.18, 3.61)) and depressive symptoms ($\hat{\beta }$ (95% CI) = 1.97 (0.007, 3.88)). An increase in sedentary time was associated with higher stress ($\hat{\beta }$(95% CI) = 0.44 (0.05, 0.83)), distress ($\hat{\beta }$(95% CI) = 14.15 (4.16, 24.15)), anxiety symptoms ($\hat{\beta }$(95% CI) = 2.29 (0.20, 4.38)) and depressive symptoms ($\hat{\beta }$(95% CI) = 2.65 (0.35, 4.95)) compared to stable sedentary time. A decrease in sedentary time was associated with higher stress (β(95% CI) = 0.32 (0.26, 0.92)) compared to stable sedentary time. Compared to participants who report no change in sleep duration, participants reporting increased sleep duration present higher stress ($\hat{\beta }$(95% CI) = 0.55 (0.18, 0.91)), distress ($\hat{\beta }$(95% CI) = 13.09 (3.34, 22.84)) and depressive symptoms ($\hat{\beta }$ (95% CI) = 2.53 (0.35, 4.72)). Similarly, a decrease in sleep duration was associated with higher stress ($\hat{\beta }$(95% CI) = 0.72 (0.33, 1.11)), distress ($\hat{\beta }$(95% CI) = 13.58 (3.31, 23.85)), anxiety symptoms ($\hat{\beta }$(95% CI) = 2.29 (0.20, 4.38)) and depressive symptoms ($\hat{\beta }$(95% CI) = 2.65 (0.35, 4.95)) compared to those reporting no change in sleep duration. Changes in MVPA, sedentary time or sleep duration, whether it increased or decreased, were not associated with resilience. 

Sensitivity analyses were conducted, assessing only those who were immunocompromised (*n* = 108), whereby all the same associations were observed between changes in lifestyle behaviors and mental health indictors, as seen in Appendix A. The only difference observed was that an increase in sleep duration was positively associated with anxiety symptoms ($\hat{\beta }$(95% CI) = 2.43 (0.09, 4.79)), whereby in the main analyses, this finding was not significant ($\hat{\beta }$(95% CI) = 1.70 (−0.28, 3.69)). These sensitivity analyses depict that after excluding the relatives of participants, changes in lifestyle behaviors are still associated with mental health indicators. 

## 4. Discussion 

This study aimed to assess the association between changes in the lifestyle behaviors of MVPA, sedentary time and sleep duration on mental health indicators during the COVID-19 pandemic lockdown among immunosuppressed individuals and their relatives. Our results indicate that unhealthy changes in lifestyle behaviors are associated with poor mental health outcomes. Specifically, decreases in MVPA are associated with higher stress, anxiety and depressive symptoms whereas increased in sedentary time is associated higher stress, distress, anxiety and depressive symptoms. Changes in sleep duration, whether it is an increased or a decrease, were associated with negative mental health outcomes: higher stress, distress and depressive symptoms. A similar trend was observed in our sensitivity analyses, after excluding the relatives of participants. Based on these findings among an understudied subgroup of immunosuppressed individuals, lifestyle behavior changes during stressful life events such as rapid outbreaks of infectious diseases have strong associations with mental health.

Extensive research supports the positive association between physical activity and mental health, even during the COVID-19 pandemic, where decreased physical activity led to worsening of mental health [55,56]. Studies investigating the associations between lifestyle behaviors and mental health suggest that during the compulsory confinements, individuals were less physically active and had more depression, anxiety and stress [11,12,57]. The current results similarly suggest that individuals who decreased their physical activity had higher distress and depressive symptoms. It is likely that physical and social distancing measures imposed a structural barrier to maintaining physical active including transportation and daily errands and this change in behavior was associated with poorer mental health outcomes. 

To support the current findings that sedentary behavior increased and was related to worsened mental health indicators, a recent meta-analysis and systematic review observed worse global mental health, depression, anxiety and quality of life across the age span with increased sedentary time over the COVID-19 pandemic [58,59,60]. Like physical activity decreases, an increase in sedentary behavior during the pandemic was inevitable due to public health confinement requirements. With the known negative associations of sedentary time and mental health [11,61], it was only expected that an increase in sedentary time would negatively influence mental health parameters. 

Our results are in line with previous studies suggesting that the COVID-19 pandemic had negative effects on sleep duration. A systematic review and meta-analysis examining sleep problems during the pandemic found that 35.7% of all populations had more sleep problems during the first eight months of the pandemic [62]. Significant sleep disturbances were reported during the pandemic across different populations such as adolescents and children [63], university students [64] and nurses [65]. To our knowledge, our findings are the first to suggest that not only were there changes in sleep during the pandemic, but that these changes significantly affected mental health status in immunosuppressed populations and their relatives. Resilience was not seen to be associated with any change in lifestyle behavior in our population. Though it has been observed that overall resilience was lower amongst adults in the USA during the COVID-19 pandemic [66,67], our results suggested that resilience was not significantly associated with lifestyle changes. 

There is evidence to suggest that the increased mental health indicators reported among immunosuppressed individuals may be due to an increased fear of contracting COVID-19 and the increased adherence to COVID-19 preventative behaviors during the mandatory stay at home order [68,69]. However, this study provides an important contribution to the limited evidence examining whether lifestyle behaviors change in physical activity, sleep and sedentary behavior relate to mental health among the immunosuppressed population and their relatives. Indeed, healthier lifestyle choices such as increased physical activity, less sedentary time and adequate sleep time are strong indicators of mental health [11,70] and such associations extend to the immunosuppressed population. An individual who is already vulnerable due to their health status may face more stress in anticipation of an increased risk of illness or infection, not only during the pandemic [71,72], but in other time periods of restraint as well. Our results suggest that a healthy lifestyle, by increasing physical activity, decreasing sedentary time and maintaining adequate sleep levels, could help mitigate the negative impact of the numerous stressful events in life that this particularly vulnerable population may already be facing. 

Limitations of this study include the cross-sectional design which precludes inference of causality. Pre-COVID lifestyle behaviors measurements are subject to recall bias which could results in misclassification of change in MVPA, sedentary time and sleep duration. The use of a self-reported questionnaire is also subject to misclassification. Additionally, the use of a convenient sample may limit the generalizability of the findings outside the scope of immunocompromised individuals. Moreover, the 10-min cut-offs used for sedentary behavior and sleep duration may be underestimating change in our population. Though the brief measurement of MVPA has previously been used before [27], the measurement of MVPA as days/week rather than minutes/day poses itself as a limitation as we are unable to detect the time per day spent doing MVPA. Finally, we acknowledge that the estimated associations between change in lifestyle behaviors and mental health indicators could be affected by residual confounding.

## 5. Conclusions

The current findings indicate that the lifestyle behavior changes during a stressful life event, such as the COVID-19 pandemic, were associated with negative mental health indicators in immunosuppressed populations. Assessing such a vulnerable population is crucial considering the stronger confinement requirements for such a population and their risk for health complications in stressful experiences. Future studies should investigate mental health indicators, including positive aspects such as resilience, in transplanted and immunosuppressed populations in various stressful life contexts.

## Figures and Tables

**Table 1 ijerph-20-02099-t001:** Participant Characteristics (*n* = 132) ^a^.

Participant Characteristics	*n* (%) or (mean ± SD)
Sex, (female)	86 (65)
Age, *n* (%)	
Under 18 years old	5 (4)
18 to 34 years old	20 (15)
35 to 54 years old	52 (39)
55 to 69 years old	51 (39)
70 years old or over	4 (3)
Transplant Status, *n* (%)	
A transplant, tissue or stem cell recipient	100 (76)
An organ, tissue or stem cell donor	4 (3)
A family member/relative of recipient or donor	24 (18)
Immunosuppressed (other reasons)	4 (3)
Medication Usage, *n* (%)	
No	105 (80)
Yes	26 (20)
Prefer not to answer	1 (1)
Employment, *n* (%)	
Employed full time	41 (31)
Employed part time	13 (10)
Self-employed	9 (7)
Unemployed	7 (5)
Retired	25 (19)
On temporary leave (illness, work related accident, maternity-paternity)	13 (10)
Student	10 (8)
Other	13 (10)
Average change in MVPA, (days/week) ^a^	−1.0 ± 3.1
Change in MVPA, *n* (%)	
Decrease	43 (33)
Stable	63 (48)
Increase	26 (19)
Average change in sedentary time, (min/day) ^a^	106.8 ± 131.6
Change in sedentary time categories, *n* (%)	
Decrease	12 (9)
Stable	30 (23)
Increase	90 (68)
Average change in sleep, (min/day) ^a^	−1.4 ± 80.7
Change in sleep duration categories, *n* (%)	
Decrease	44 (33)
Stable	37 (28)
Increase	51 (39)
Stress, (mean ± SD)	3.4 ± 0.9
Anxiety Symptoms, (mean ± SD)	5.6 ± 4.7
Distress, (mean ± SD)	32.7 ± 23.8
Depressive Symptoms, (mean ± SD)	6.4 ± 5.2
Resilience, (mean ± SD)	21.8 ± 5.0

^a^ Values are unadjusted means ± standard deviations.

**Table 2 ijerph-20-02099-t002:** Mental health outcomes according to lifestyle behavior change (*n* = 132) ^a^.

**Moderate to Vigorous Physical Activity**	**Decrease** **(*n* = 43)**	**Stable** **(*n* = 63)**	**Increase** **(*n* = 26)**
Stress	3.5 ± 0.9	3.1 ± 0.9	3.2 ± 1.0
Distress	37.4 ± 24.6	28.9 ± 23.1	29.6 ± 22.3
Resilience Scale	21.4 ± 5.2	22.0 ± 4.7	22.4 ± 5.1
Anxiety Symptoms	6.7 ± 5.3	4.8 ± 4.2	4.9 ± 3.9
Depression Symptoms	7.4 ± 5.6	5.5 ± 4.7	5.6 ± 5.1
**Sedentary Time**	**Decrease** **(*n* = 12)**	**Stable** **(*n* = 30)**	**Increase** **(*n* = 90)**
Stress	3.6 ± 0.9	3.2 ± 0.8	3.5 ± 1.0
Distress	29.2 ± 20.68	22.2 ± 21.3	36.7 ± 24.0
Resilience Scale	21.1 ± 5.74	22.8 ± 5.1	21.6 ± 4.8
Anxiety Symptoms	5.5 ± 3.37	3.6 ± 3.6	6.3 ± 5.0
Depression Symptoms	6.0 ± 4.86	4.3 ± 4.6	7.1 ± 5.3
**Sleep Duration**	**Decrease** **(*n* = 44)**	**Stable** **(*n* = 37)**	**Increase** **(*n* = 51)**
Stress	3.7 ± 0.9	3.0 ± 0.7	3.5 ± 1.1
Distress	38.5 ± 23.9	21.6 ± 18.0	35.8 ± 25.1
Resilience Scale	21.6 ± 4.9	22.8 ± 5.5	21.3 ± 4.7
Anxiety Symptoms	6.8 ± 4.5	3.8 ± 3.6	5.9 ± 5.3
Depression Symptoms	7.5 ± 5.5	4.0 ± 3.8	7.0 ± 5.4

^a^ Values are unadjusted means ± standard deviations.

**Table 3 ijerph-20-02099-t003:** Associations between changes in lifestyle behaviors on mental health indicators (*n* = 132).

	Stress			Distress			Resilience			Anxiety Symptoms			Depressive Symptoms		
	$\hat{\beta }$	95% CI	*p* ^a^	$\hat{\beta }$	95% CI	*p* ^a^	$\hat{\beta }$	95% CI	*p* ^a^	$\hat{\beta }$	95% CI	*p* ^a^	$\hat{\beta }$	95% CI	*p* ^a^
MVPA ^a^															
Stable (*n* = 63)	Ref			Ref			Ref			Ref			Ref		
Increase (*n* = 26)	−0.17	0.58, 0.24	0.41	3.14	−7.60, 13.89	0.56	0.29	−2.17, 2.78	0.81	0.54	−1.56, 2.74	0.59	0.41	−1.96, 2.80	0.73
Decrease (*n* = 43)	0.20	0.13, 0.53	0.22	**9.09**	**0.52, 17.65**	**0.04**	−0.62	−2.59, 1.35	0.53	**1.89**	**0.18, 3.61**	**0.03**	**1.97**	**0.07, 3.88**	**0.04**
Sedentary time															
Stable (*n* = 30)	Ref			Ref			Ref			Ref			Ref		
Increase (*n* = 90)	**0.44**	**0.05, 0.83**	**0.03**	**14.15**	**4.16, 24.15**	**0.005**	0.97	−3.30, 1.37	0.41	**2.37**	**0.34, 4.40**	**0.02**	**2.39**	**0.13, 4.65**	**0.03**
Decrease (*n* = 12)	**0.32**	**0.26, 0.92**	**0.02**	1.98	−13.10, 17.06	0.79	−1.44	−4.97, 2.08	0.42	0.66	−2.40, 3.72	0.67	0.49	−2.92, 3.90	0.77
Sleep duration															
Stable (*n* = 37)	Ref			Ref			Ref			Ref			Ref		
Increase (*n* = 51)	**0.55**	**0.18, 0.91**	**0.003**	**13.09**	**3.34, 22.84**	**0.009**	−1.29	−3.57, 0.97	0.26	1.70	−0.28, 3.69	0.09	**2.53**	**0.35, 4.72**	**0.02**
Decrease (*n* = 44)	**0.72**	**0.33, 1.11**	**<0.001**	**13.58**	**3.31, 23.85**	**0.009**	−0.94	−3.33, 1.45	0.44	**2.29**	**0.20, 4.38**	**0.03**	**2.65**	**0.35, 4.95**	**0.02**

^a^ *p* value adjusted for age, sex and physical illness diagnosis. ^b^ MVPA = Moderate-to-vigorous physical activity. Bold indicate statistically significant results at *p* < 0.05.

## Data Availability

The datasets used and/or analyzed during the current study are available from the corresponding author on reasonable request.

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
