# Peer review of "The Association between Change in Lifestyle Behaviors and Mental Health Indicators in Immunosuppressed Individuals during the COVID-19 Pandemic"

_ijerph, 2023, doi:10.3390/ijerph20032099_

Round 1

Reviewer 1 Report

what is a healthy lifestyle?  A healthy lifestyle is a way of living that lowers the risk of being seriously ill or dying early. It includes a wide range of things such as eating habits, social habits, etc. In light of this, the research has to clarify what a healthy lifestyle is before identifying the type of lifestyle it will be focusing on (physical activity as a healthy lifestyle).

Line 32 introduces unhealthy lifestyles. That is fine but you also need to explicitly introduce healthy lifestyles. What is it that amounts to healthy lifestyles? (Apparently, lines 331-333 did this momentarily. This must also be done in the background section).

Why did they study choose to focus on immunosuppressed population’s mental health? What are they? What makes them noteworthy?

Line 56 – why are immunosuppressed individuals at an increased risk of severe complications? You may need to explain this in detail.  

Line 61-65 – please cite at least 3 articles to support your assertion.

The background lacks a logical flow. Where is paragraph 4 coming from? Line 78-89. Where does it fit? Try to find a way of linking it to the rest of the background section.

study design - This study uses data from the COVID-Immuno Study –  I am a bit confused here. I thought the researchers collected primary data. Who collected the data? The entity/persons that collected the data should be mentioned here.

Data collection - being transplanted and immunosuppressed, being immunosuppressed – how did you identify the immunosuppressed individuals?

The estimation techniques need to be discussed in detail. The study only mentions in passing that it used linear regression. Describe the reasons this method was chosen. What were the characteristics of the data that led to the application of linear regression.

Author Response

We thank the reviewers whose comments and suggestions helped improve the manuscript. Responses to reviewers’ comments and changes in the manuscript are presented below. We have used track changes to portray the edits and additions to the manuscript.

  1. what is a healthy lifestyle?  A healthy lifestyle is a way of living that lowers the risk of being seriously ill or dying early. It includes a wide range of things such as eating habits, social habits, etc. In light of this, the research has to clarify what a healthy lifestyle is before identifying the type of lifestyle it will be focusing on (physical activity as a healthy lifestyle).

We thank the reviewer for their suggestion. We have now included a definition for healthy lifestyles on page 1, lines 39-40. “A healthy lifestyle includes a range of activities to encourage habits that reduces the risk of morbidity and increases life expectancy.”

  1. Line 32 introduces unhealthy lifestyles. That is fine but you also need to explicitly introduce healthy lifestyles. What is it that amounts to healthy lifestyles? (Apparently, lines 331-333 did this momentarily. This must also be done in the background section).

This has been implemented on page 1, lines 40-41. “Such a healthy lifestyle is encompassed by a plethora of habits such as but not limited to, healthy eating, socialization, regular physical activity, and adequate sleep.”

  1. Why did they study choose to focus on immunosuppressed population’s mental health? What are they? What makes them noteworthy?

This study chose to focus on immunosuppressed populations mental health due to the lack of research in this important and crucial area among this specific population. Since immunosuppressed populations are living in constant fear of contracting illnesses, we were curious to see how simple healthy lifestyle measures influence their mental health status. What makes this population noteworthy is that it is a vulnerable one, who should be allocated extra care and support. As such, we have added sentences in the introduction to better explain this on page 2, lines 67-69, and 104-106.

Individuals who are immunocompromised are more likely to contract illnesses or be ill for extended periods of times due to their weakened immune system.”

“Considering the high level of stress immunosuppressed individuals have while avoiding illness, mental health is crucial to assess in a vulnerable population.”

  1. Line 56 – why are immunosuppressed individuals at an increased risk of severe complications? You may need to explain this in detail.  

An explanation for this sentence has been added to page 2, lines 67-69.

Individuals who are immunocompromised are more likely to contract illnesses or be ill for extended periods of times due to their weakened immune system.”

  1. Line 61-65 – please cite at least 3 articles to support your assertion.

2 additional citations have been added to the manuscript on line 74.

Whitley E, Reeve K, Benzeval M: Tracking the mental health of home-carers during the first COVID-19 national lockdown: evidence from a nationally representative UK survey. Psychol Med 2021:1-10.

Li Q, Zhang H, Zhang M, Li T, Ma W, An C, Chen Y, Liu S, Kuang W, Yu X, Wang H: Prevalence and Risk Factors of Anxiety, Depression, and Sleep Problems Among Caregivers of People Living With Neurocognitive Disorders During the COVID-19 Pandemic. Front Psychiatry 2020, 11:590343.

  1. The background lacks a logical flow. Where is paragraph 4 coming from? Line 78-89. Where does it fit? Try to find a way of linking it to the rest of the background section.

Thank you for your suggestion, these sentences have now been linked to the previous paragraph describing physical activity and sedentary behavior from lines 82-84.

  1. study design - This study uses data from the COVID-Immuno Study –  I am a bit confused here. I thought the researchers collected primary data. Who collected the data? The entity/persons that collected the data should be mentioned here.

We have reworded the study design to include that our team has conducted the study on page 3, line 163. Co-Authors MD and ID are co-PIs of this study.

Our team conducted the COVID-Immuno Study…”

  1. Data collection - being transplanted and immunosuppressed, being immunosuppressed – how did you identify the immunosuppressed individuals?

The specific question used to identify immunosuppressed individuals has been implemented in the manuscript on page 3, lines 189-194.

“In specific, participants were asked to identify if they were either, i) a transplant, tissue or stem cell recipient; ii) a family member/relative of a transplant/tissue/stem cell recipient or donor; iii) an organ, tissue or stem cell donor; iv) none of the above; v) prefer not to answer. Individuals who selected “none of the above” had an option of explaining if they were immunosuppressed for other reasons, but not transplanted.”

  1. The estimation techniques need to be discussed in detail. The study only mentions in passing that it used linear regression. Describe the reasons this method was chosen. What were the characteristics of the data that led to the application of linear regression.

A succinct explanation of the method has been added to page 5 lines 312-317.

“Multivariate linear regressions were used to assess whether an increase or decrease in lifestyle behaviors of MVPA, sedentary time and sleep duration is associated with each mental health indicators of stress, distress, resilience, anxiety, and depressive symptoms compared to no changes. Linear regressions were chosen to estimate the association between the categorical exposure of lifestyle behavior changes and continuous outcome of mental health indicators”

Reviewer 2 Report

Thank you for the opportunity to review the manuscript titled The Association between Change in Lifestyle Behaviours and Mental Health Indicators in Immunosuppressed Individuals during the COVID-19 Pandemic.  This research study examines the relationship between sedentary lifestyle variables and mental health outcomes for individuals in the immunosuppressed population.  The authors conclude that this population is at risk for negative mental health outcomes based on unhealthy lifestyle changes during the COVID-19 pandemic.

The manuscript is well-written, and the methodology is well-explained.  Results are clearly presented, and the conclusions are directly related to the analysis.  I have no recommendations for improvement regarding the technical aspects of this paper.

Three minor corrections I would suggest are:

Line 46:  Change the words "faced in" to wither "placed in" or "faced with"

Line 67:  Place a comma after between the words " challenge which"

Line 122: Change the words "such as" to "including."  Using the words such as indicates that there are other demographic variables that the authors have not listed, which would be inappropriate when reporting the methodology.

Author Response

We thank the reviewer for their comments.

  1. Line 46:  Change the words "faced in" to wither "placed in" or "faced with"

Thank you for pointing that out. We have updated the manuscript accordingly with “faced with” on line 46.

  1. Line 67:  Place a comma after between the words " challenge which"

Thank you for pointing that out. We have updated the manuscript accordingly on line 78

  1. Line 122: Change the words "such as" to "including."  Using the words such as indicates that there are other demographic variables that the authors have not listed, which would be inappropriate when reporting the methodology.

We have updated the manuscript accordingly on line 184.

Reviewer 3 Report

The manuscript provides the findings of a study conducted with objective to assess the association between changes in the lifestyle behaviours of moderate-to-vigorous physical activity, sedentary time and sleep duration on mental health indicators during the COVID-19 pandemic lockdown among immunosuppressed individuals and their relatives.  The study addresses a current and relevant issue in the field of health public. The manuscript is properly structured, the theoretical basis is satisfactory, the references used are current and relevant to the subject of the study. However, with the intention of helping to improve the manuscript, I present some items that could possibly be adjusted.

(1) Methods - 2.5. Lifestyle behaviours

Moderate-to-vigorous physical activity (MVPA) – Lines 166 to 177: If the International Physical Activity Questionnaire (IPAQ) was used to measure the physical activity of the study participants, I don’t understand why only the number of days/week spent in each of the two intensities was observed in the data analysis. This procedure is uncommon. I suggest measuring moderate-to-vigorous physical activity by calculating the number of days/week multiplied by the amount of minute/day.

(2) Methods - 2.5. Lifestyle behaviours

Sedentary time – Line 180: The term "night" should be deleted (… on a typical weekday night and on a weekend day separately both on a ….).

(3) Methods - 2.5. Lifestyle behaviours

Sedentary time – Lines 186 to 188: I believe that using a 10-minute bout of change for sedentary time based on changes in MVPA bouts [previous literature 43] is an inadequate criterion. On the same day, the possibility of variation in the sedentary time is much greater than of MVPA; therefore, the bout of chance for sedentary time should also be higher.

(4) Methods - 2.5. Lifestyle behaviours

Sleep duration – Lines 189 to 198: I also believe that using a 10-minute bout of change for sleep duration is an inadequate criterion. In the same day/night, the possibility of variation in the duration of sleep/night is much greater than the MVPA/day (approximately 7-8 hours/night versus 1-2 hours/week); therefore, the bout of chance for sleep duration should also be higher.

(5) Results – Table 2 - Mental health outcomes according to lifestyle behaviour change – Line 247

I suggest adding results from an analysis of covariance (p value adjusted for potential confounders - age, sex, and physical illness diagnosis) to compare each mental health indicator (stress, distress, resilience, anxiety and depression symptoms) according to lifestyle behaviour (MVPA, sedentary time, and sleep duration) change categories (increase, decrease, stable).  

Author Response

We thank the reviewer for their comments and suggestions.

  1. Methods - 2.5. Lifestyle behaviours. Moderate-to-vigorous physical activity (MVPA) – Lines 166 to 177: If the International Physical Activity Questionnaire (IPAQ) was used to measure the physical activity of the study participants, I don’t understand why only the number of days/week spent in each of the two intensities was observed in the data analysis. This procedure is uncommon. I suggest measuring moderate-to-vigorous physical activity by calculating the number of days/week multiplied by the amount of minute/day.

Although we agree with the reviewer that measuring physical activity by minutes/day is crucial, minutes of activity was not collected and could not be calculated and reported. We have modified the manuscript and included this as a limitation on page 10, lines 490-492

“Though the brief measurement of MVPA has previously been used before [27], the measurement of MVPA as days/week rather than minutes/day poses itself as a limitation as we are unable to detect the time per day spent doing MVPA.”

  1. Methods - 2.5. Lifestyle behaviours

Sedentary time – Line 180: The term "night" should be deleted (… on a typical weekday night and on a weekend day separately both on a ….).

Thank you for pointing that out, this has now been adjusted accordingly, on lines 260 and 270.

  1. Methods - 2.5. Lifestyle behaviours

Sedentary time – Lines 186 to 188: I believe that using a 10-minute bout of change for sedentary time based on changes in MVPA bouts [previous literature 43] is an inadequate criterion. On the same day, the possibility of variation in the sedentary time is much greater than of MVPA; therefore, the bout of chance for sedentary time should also be higher.

As we do not have the criterion numbers, we felt that being consistent with MVPA was appropriate. We wanted to capture perceived change, and not the specific amount of change and 10 minutes is a viable intervention target to start and is tangible to immunosuppressed individuals. We have modified the manuscript to include this as a limitation underestimating change on page 9, lines 488-490.

“Moreover, the 10-minute cut-offs used for sedentary behavior and sleep duration may be underestimating change in our population.”

  1. Methods - 2.5. Lifestyle behaviours

Sleep duration – Lines 189 to 198: I also believe that using a 10-minute bout of change for sleep duration is an inadequate criterion. In the same day/night, the possibility of variation in the duration of sleep/night is much greater than the MVPA/day (approximately 7-8 hours/night versus 1-2 hours/week); therefore, the bout of chance for sleep duration should also be higher.

Please see our response to the previous comment (3), which addresses this comment as well.

  1. Results – Table 2 - Mental health outcomes according to lifestyle behaviour change – Line 247

I suggest adding results from an analysis of covariance (p value adjusted for potential confounders - age, sex, and physical illness diagnosis) to compare each mental health indicator (stress, distress, resilience, anxiety and depression symptoms) according to lifestyle behaviour (MVPA, sedentary time, and sleep duration) change categories (increase, decrease, stable).  

Such results are included in Table 3 and are described in page 7. The formatting, with supplementary tables included throughout the manuscript, may have distracted the reviewer from this analysis. We have removed the supplementary tables from the main manuscript and have them in a separate document. 

Round 2

Reviewer 1 Report

n/a